# Kinetics of Carbon Enrichment in Austenite during Partitioning Stage Studied via In-Situ Synchrotron XRD

**DOI:** 10.3390/ma16041557

**Published:** 2023-02-13

**Authors:** Farnoosh Forouzan, Roohallah Surki Aliabad, Ali Hedayati, Nazanin Hosseini, Emad Maawad, Núria Blasco, Esa Vuorinen

**Affiliations:** 1Department of Engineering Sciences and Mathematics, Luleå University of Technology, SE-97187 Luleå, Sweden; 2Materials and Mechanical Engineering, University of Oulu, 90014 Oulu, Finland; 3Helmholtz-Zentrum Hereon, Institute of Materials Physics, Max-Planck-Straße 1, 21502 Geesthacht, Germany

**Keywords:** in-situ synchrotron XRD, high-resolution dilatometry, quenching and partitioning (Q&P), high-carbon steel, advanced high strength steels, martensitic/bainitic phase transformation

## Abstract

The present study reveals the microstructural evolution and corresponding mechanisms occurring during different stages of quenching and partitioning (Q&P) conducted on 0.6C-1.5Si steel using in-situ High Energy X-Ray Diffraction (HEXRD) and high-resolution dilatometry methods. The results support that the symmetry of ferrite is not cubic when first formed since it is fully supersaturated with carbon at the early stages of partitioning. Moreover, by increasing partitioning temperature, the dominant carbon source for austenite enrichment changes from ongoing bainitic ferrite transformation during the partitioning stage to initial martensite formed in the quenching stage. At low partitioning temperatures, a bimodal distribution of low- and high-carbon austenite, 0.6 and 1.9 wt.% carbon, is detected. At higher temperatures, a better distribution of carbon occurs, approaching full homogenization. An initial martensite content of around 11.5 wt.% after partitioning at 280 °C via bainitic ferrite transformation results in higher carbon enrichment of austenite and increased retained austenite amount by approximately 4% in comparison with partitioning at 500 °C. In comparison with austempering heat treatment with no prior martensite, the presence of initial martensite in the Q&P microstructure accelerates the subsequent low-temperature bainitic transformation.

## 1. Introduction

The quenching and partitioning (Q&P) heat treatment method has been investigated in both academia and industry during the past two decades due to the good strength and formability of the resulting products [1]. It consists of partial martensite formation from a completely or partially austenitic microstructure, followed by heating at the same or higher temperature to facilitate carbon partitioning and stabilization of the retained austenite [2]. In this process, two main mechanisms have been proposed for carbon partitioning of the untransformed austenite: (i) diffusion of carbon from carbon-supersaturated martensite; and (ii) carbon enrichment of austenite associated with the formation of carbide-free bainite [3].

Both mechanisms lead to the formation of local high-carbon austenite ($\gamma_{HC}$) regions, austenite films between the plates of bainitic ferrite blocks, trapped among the sheaves of bainite [4], and austenite regions adjacent to tempered martensite [5]. However, as stated by Guo et al. [6], the latter mechanism results in a bimodal carbon distribution in austenite grains having carbon concentrations of 0.58 and 1.12 wt.% for low- and high-carbon austenite respectively.

Depending on chemical composition, the minimum carbon concentration for stabilizing austenite at room temperature varies; for example, Hyughe et al. [7] reported a minimum of 0.8 wt.% carbon for 0.2C-1.41Si-2.31Mn steel, while 1.15 wt.% was needed in another steel grade reported by Maheswari et al. [8].

However, competing processes like carbide precipitation and austenite decomposition must be minimized to ensure a successful partitioning process. This is accomplished by delaying the development of carbides using alloying elements such as Si, Al, or P [9] and optimizing Q&P heat treatment [10,11]. Unlike the effect on cementite formation, the effect of these alloying elements on transitional carbides (ε and η) is not very clear, as some transitional carbides can be found in lower-carbon sheet steels treated by Q&P [12]. Therefore, understanding the mechanisms and conditions responsible for maintaining the stability of retained austenite (RA) at quenching temperature (QT) and holding at the subsequent partitioning stage is critical for the development of Q&P steels.

The Q&P heat treatment has been extensively investigated in the past [2,9,13,14,15,16]. Typical characterization techniques (e.g., XRD, SEM, TEM, and Atom Probe Tomography (APT)) have been widely used to investigate microstructural development. However, metallurgical processes involving carbon, such as diffusion and partitioning or carbide precipitation, are challenging to evaluate using traditional methods. Therefore, high acquisition rate techniques are of an immense importance in investigation of these phenomena.

Past research on steels has shown that the use of in-situ approaches to gather time-resolved information is useful [17,18,19,20,21,22]. Furthermore, recent studies on microstructural development during Q&P treatment suggest that in-situ HEXRD is among the best approaches for studying carbon partitioning [7,23,24,25,26,27,28,29].

Understanding the microstructural evolution involved in the Q&P process requires obtaining information on carbon diffusion and distribution in the different constituents included in the microstructure, i.e., retained austenite, bainitic ferrite, and carbides. In particular, in-situ HEXRD and dilatometry, combined with extremely high acquisition rates, have enabled researchers [7,26,30,31,32] to gain a better understanding of austenite evolution throughout the entire Q&P process. Furthermore, the parameters that affect the kinetics of the process can also be elaborately investigated only with in-situ methods. Some theories claim that the existence of prior martensite (PM) accelerates the bainite transformation [33,34,35,36]. However, as the competing phenomena (e.g., carbide formation, carbon trapping at dislocations, and decomposition of austenite to bainite) occur simultaneously, the mechanisms need to be further investigated.

Therefore, the objective of this study is to increase knowledge about the influence of partitioning temperature on the microstructural evolution of high-carbon high-silicon steel, and about the partitioning mechanisms that lead to austenite stabilization during different stages of the Q&P treatment. These include: (a) partial austenite to martensite transformation during the initial quench, (b) the reheating stage to the partitioning temperature, (c) carbon enrichment and carbon homogenization during partitioning, and (d) the stability of retained austenite at room temperature. For this purpose, high partitioning temperatures were applied for short periods, and these were compared with lower partitioning temperatures utilized for longer periods. Additionally, in this study, particular attention was devoted to the effect of three partitioning temperatures on partitioning mechanisms (corresponding to different fractions of bainitic ferrite formed during the partitioning). Furthermore, the effect of pre-existing martensite on the kinetics of bainite transformation was investigated.

## 2. Materials and Methods

### 2.1. Material and Thermal Treatments

The chemical composition of the studied steel (denoted as 06CV), produced by ASCOMETAL, France, is Fe-0.6C-1.25Mn-1.6Si-1.75Cr-0.15Mo-0.12V (in wt.%). Four specimens from a hot-rolled and annealed bar with an initial pearlitic microstructure were machined into hollow cylinders with 4 mm external diameter, 1 mm wall thickness, and 10 mm length (L_0_). It was assumed that the effect of texture would be negligible in this study since this was confirmed by Electron Backscatter Diffraction (EBSD) in [37]. All heat treatments started with complete austenitization of the materials, followed by quenching with pressured Argon gas to 165 °C and hold at temperature for 13 s, except for one sample which was isothermally austempered at 280 °C for 30 min. As shown in Figure 1, the procedure included direct quenching of one sample to 30 °C while heating the others to 280, 400, and 500 °C for various holding durations.

### 2.2. High Energy X-ray Diffraction (HEXRD)

In-situ high-energy XRD measurements were carried out using transmission geometry at the P07B beamline of Hereon at PETRA III, DESY, Germany. The experiments were performed using monochromatic synchrotron X-ray radiation with a photon energy of 87.1 keV (λ = 0.14235 Å) and a beam size of 0.7 × 0.7 mm^2^ to achieve a short acquisition time, reasonable grain statistics, and reasonably good angular resolution. Instrument calibration was carried out to determine the values required for X-ray data analysis (e.g., line broadening, wavelength, detector non-orthogonality, and distance between the sample and the detector) using LaB_6_ powder (SRM 660C NIST, USA). Diffraction patterns were obtained continuously during the experiments with a Perkin Elmer (USA) XRD1621 X-ray detector with a pixel size of 200 µm positioned at a distance of 1392 mm behind the sample. This configuration resulted in the detection of full Debye–Scherrer rings up to a maximum 2Theta angle of 11.2°. Two modes were employed in diffraction pattern recording: a slow mode used during austenitizing and at the end of the heat treatments, with an exposure time of 3.2 s for each frame, and a fast mode with an exposure time of 0.3 s for the quenching section and the beginning of partitioning (the area is highlighted in Figure 1).

### 2.3. Dilatometry

The thermal treatments were performed with a dilatometer DIL 805A/D from TA Instruments, USA, located in the beam, and a Pt/Pt–Rh thermocouple centrally spot-welded at the surface of the specimen close to the X-ray illuminated volume. It should be noted that as the thermal treatments before the first quenching were the same for all experiments, the average of four measurements is reported in the following sections.

### 2.4. Rietveld Refinement Analysis

#### 2.4.1. Debye–Scherrer Ring Integration

Integration of the recorded Debye–Scherrer rings was carried out azimuthally using the MAUD program with a 5° step corresponding to 72 patterns per image. The obtained 1D diffraction data were then processed using a Rietveld refinement procedure implemented with MAUD software.

#### 2.4.2. Instrument Calibration

The instrumental line-broadening contribution was determined by using the LaB_6_ powder sample. This can be achieved in MAUD by setting the line-broadening model to none and the size–strain model to isotropic for this LaB_6_ phase, and refining the Caglioti, asymmetry, and Gaussian parameters for the instrument object (for more details, see ref. [38]). Subsequently, for other specimens, these determined instrumental parameters were used as fixed, and only background, scale, basic phase, and crystal structure parameters were refined. The weighted profile R-factor (R_wp_), the expected R-factor (R_exp_), and chi squared (χ^2^) were used as indicators of the quality of fitting for each refinement, where R_wp_ > R_exp_ and 1 < χ^2^ < 3.5. Moreover, all the fittings were also checked graphically by comparing the detected patterns to simulated ones to ensure that the model was plausible [39].

## 3. Results

### 3.1. Coefficient of Thermal Expansion (CTE)

The contribution of thermal expansion to the lattice parameter can be calculated using the linear range Ln (a/a_0_) as a function of temperature, as shown in Figure 2, which corresponds to a cooling step from 910 to 240 °C. According to the dilatometry and XRD data, the alloy is fully austenitic in this region. The decrease in the austenite lattice parameter with temperature can therefore be attributed to austenite’s thermal contraction [23].

According to Lu et al. [40], the molar volume of nonmagnetic phases can be calculated using the following equation:(1)$$V_{m}^{nonmag}\left(T\right)=V_{0}exp\left(\int_{T_{0}}^{T}3\alpha dT\right)$$
where *V*_0_ is the molar volume at the reference temperature *T*_0_. α denotes the CTE of a phase in the nonmagnetic state, reported as an independent thermal coefficient in the case of Fe-fcc. The molar volume (*V_m_* (*T*)) is directly related to $a_{\gamma }^{3}$. Thus, the influence of temperature on the lattice parameter of the austenite is determined by Equation (2):(2)$$a_{\gamma }^{T}=a_{\gamma }^{0}exp\left(\alpha \;\left(T-T_{0}\right)\right)$$

Consequently, the coefficient of thermal expansion (CTE) of the 06CV steel is calculated as 2.2641 × 10^−5^ K^−1^, which is consistent with previous studies [6,40,41,42,43] showing CTEs in the range of 2.0–2.5 (×10^−5^ K^−1^) independent of the initial microstructure and chemical composition.

### 3.2. Austenite Carbon Content Estimation

Different empirical equations for estimating the austenite lattice parameter at room temperature based on chemical composition have been proposed in the literature [17,44,45,46,47,48]. In all of them, the weight of the carbon content is much greater than that of the substitutional elements. In this study, the Equation (3) proposed by Dyson and Holmes [41] was used:(3)$$a_{\gamma }=3.5847+0.0330\times \left[C\right]+0.00095\times \left[Mn\right]-0.0002\times \left[Ni\right]\;+0.0006\times \left[Cr\right]+0.0015\times \left[Cu\right]$$

In the above equation, 3.5847 Å corresponds to the austenite lattice parameter in pure iron at room temperature. Using the composition provided in Section 2.1, the austenite lattice parameter can be calculated to be 3.6067 Å.

However, the thermal expansion effect at elevated temperatures must be excluded, which can be done using the expression proposed by Denand et al. [41], Equation (4):(4)$$w_{C}^{\gamma }=\frac{\Delta a_{\gamma }}{A}+w_{C0}^{\gamma }$$
where $w_{C}^{\gamma }$ is the total carbon concentration change in the austenite, $\Delta a_{\gamma }$ is the difference in the lattice parameter of the austenite without thermal contribution over time at the same temperature, $w_{C0}^{\gamma }$ is the nominal carbon content of the steel, and *A* is the constant parameter of 0.033 wt.% extracted from the Dyson and Holmes equation.

### 3.3. First Quench (Initial Martensite State)

Austenitization was performed at 910 °C and the first quenching treatment was stopped at 165 °C. Figure 3 shows the evolution of the γ→ά transformation of the as-quenched specimen. The slope of dilation ($\frac{d\;\left(RCL\right)}{dT})$ vs. time shows two changes in the slope, indicating the first M_s_ temperature ($M_{s}^{Q1}$ = 240 °C) and the second M_s_ temperature ($M_{s}^{Q2}$) at around 185 °C. Although XRD patterns do not show any martensite peaks in stage I, the XRD data clearly confirms the second M_s_ temperature ($M_{s}^{Q2}$). Figure 3b represents a logarithmic contour graph of the XRD patterns versus time. According to this graph, when the temperature is reduced, martensite peaks appear gradually. The first doublet peaks that show up at 183.1 °C are (101) and (110), indicating that this temperature is a good estimate of $M_{s}^{Q2}$.

The authors [49] have noted that micro-segregation generates bands of enriched and deficient Mn–Cr areas, which affected the M_s_ temperature of the bands in a separate experiment on the same steel (bands with higher amounts of Cr-Mn resulted in lower M_s_ temperature, and vice versa). Therefore, it can be concluded that partial segregation affected the first quench, based on the foregoing observations and previous research on the studied steel. At $M_{s}^{Q1}$ = 240 °C, the minor bands which are depleted Mn–Cr areas began to transform into martensite, but the percentage was most likely below the synchrotron detection limit. Subsequently, the major enriched Mn–Cr regions started transforming at $M_{s}^{Q2}$ = 183 °C. Consequently, as the undercooling was around 20 °C, only 11.5 wt.% martensite was formed.

Moreover, at this temperature range (183 → 165 °C), the martensite exhibited carbon depletion as the tetragonality decreased by quenching temperature, from 1.030 at 183 °C to 1.026 at 165 °C. However, estimation of the average carbon atom diffusion in the austenite using Equation (5) resulted in a radial distance (*r*) of carbon atom movement of 0.4 nm after 13 s of holding time at 165 °C, which could not significantly change the carbon enrichment of the austenite. This also explains the constant value of the mean austenite carbon content measured by XRD Rietveld refinement analysis during this stage.
(5)$$r=2.4\sqrt{D\cdot t}$$

*D* in Equation (5) is the carbon diffusivity defined by Equation (6), with *t* indicating time in seconds:(6)$$D\left(\frac{cm^{2}}{s}\right)=\left(0.04+0.08C\right)exp^{\left(\frac{-31350}{R_{c}T}\right)}$$

In Equation (6), *R_c_* is the gas constant (1.987 $\frac{\mathrm{cal}}{k\cdot \mathrm{mol}}$), *T* is the temperature in Kelvin, and *C* is the nominal carbon content of the steel (in wt.%) [50].

### 3.4. Reheating Stage to the Partitioning Temperature

Figure 4a depicts the diffusivity of carbon into austenite and the evolution of the bct phase (tempered martensite) and the bcc phase (bainitic ferrite) during heating from the first quench temperature (165 °C) to the partitioning temperature. At partitioning temperatures lower than 400 °C, the mean austenite carbon content (pink circles) does not vary significantly by temperature. However, it increases considerably at temperatures above 400 °C. As given by Equation (5), carbon diffusivity is considerably increased as a function of higher temperature.

#### 3.4.1. FCC Phase Evolution

In addition to the mean carbon content, the full width at half maximum peak values (FWHM) of diffraction peaks can be used to evaluate chemical homogeneity/heterogeneity in the austenite. The decrease in FWHM is primarily attributed to the homogeneity of austenite carbon content, and vice versa [41]. The FWHM results in Figure 4b measured from $\gamma_{200}$ and $\gamma_{202}$ peaks, which are not overlapped by ferrite/martensite peaks [6], show a positive slope, indicating an increased heterogeneity in the austenite carbon content.

One explanation for this heterogeneity is short-range carbon atom diffusion out of tetragonal ferrite into neighboring filmy austenite [51]. Therefore, two forms of austenite, i.e., high- and low-carbon austenite, can be detected by analyzing the FWHM changes, which is consistent with prior studies [48,52] and will be discussed in detail in Section 3.6.

#### 3.4.2. Tetragonal BCT Phase

According to the XRD results shown in Figure 4a, describing the quantitative results achieved during heating from 165 °C to 500 °C, it can be seen that the tetragonality diagram shows a decrease, which indicates carbon depletion from initial martensite by increasing the temperature.

This observation confirms the previous calculations of the solubility of carbon in tetragonal ferrite in equilibrium with austenite [51]. The results of Figure 4a also confirm that the symmetry of ferrite in equilibrium with austenite is not cubic when first formed, since it is fully supersaturated with carbon [53]. It is worth noting that tetragonality in steels with C<0.6 wt.% has a direct linear relation with carbon concentration in martensite, with $\frac{c}{a}=1+0.031\times C\;\mathrm{wt}.\%$ [54].

### 3.5. Effect of Partitioning Temperature on Partitioning Mechanisms

Figure 5 depicts the microstructural evolution of bainitic ferrite and austenite during the partitioning stage at different partitioning temperatures. It demonstrates that, in addition to carbon partitioning from martensite to austenite, bainitic transformation is another active mechanism for austenite carbon enrichment. 

It is worth noting that no evidence of carbide formation was found during partitioning as a competitive carbon-consuming mechanism in this steel. The XRD images were recorded every 0.3 s, which gives a high resolution for tracking carbide formation including transitional carbides [13]. Therefore, it can be assumed that unless some amount of the carbon atoms were entrapped by the dislocations [55] or clustered [56] in bainitic ferrite, the remaining atoms will accompany the carbon partitioning. 

Figure 5a reveals that nearly 8% of new bainitic ferrite forms during the partitioning stage at 280 °C for 900 s, but the mean carbon content of the austenite only slightly increases. The FWHM of the austenite peaks, on the other hand, shows an increase, which is related to the heterogeneity of carbon in retained austenite. Because of the low carbon diffusivity at 280 °C, carbon atoms can only migrate roughly 140 nm even after 900 s of holding time. This results in the formation of two types of austenite, high- and low-carbon austenite, as shown in Figure 5, which are described in the following section. At this partitioning temperature, austenite enrichment by carbon atoms from the martensite and the bainite transformation is more consistent with the experimental results.

Figure 5b demonstrates partitioning behavior at 400 °C, where less than 1.5 wt.% bainitic ferrite is formed, while the amount of carbon partitioned into austenite is doubled compared to partitioning at 280 °C. In addition, based on the changes in FWHM for the $\gamma_{200}$ and $\gamma_{202}$ peaks, it can be concluded that carbon atoms distribute more homogenously in austenite. All of these are consistent with higher carbon diffusivity and longer possible diffusion distance, as calculated for 150 s of partitioning, r = 0.73 μm. Both carbon partitioning to austenite from martensite and bainitic ferrite formation mechanisms are active at this partitioning temperature.

In contrast, at 500 °C, carbon partitioning to austenite from martensite is the dominant mechanism. As can be seen in Figure 5c, the bainitic ferrite percentage barely increases. Nevertheless, since carbon diffusivity is highest compared with other partitioning temperatures, the mean carbon partitioned into austenite will be four times higher than at 280 °C after 5 s of partitioning, and thereafter plateaus with a slight positive slope. More significantly, as time passes, the FWHM reveals that the carbon homogeneity in the austenite increases, which is consistent with r = 1.2 μm after 20 s of partitioning.

Moreover, in Figure 5a,b, at the partitioning temperatures of 280 and 400 °C, the dilation behavior of the samples (blue curves) shows a significant correlation with the bainitic ferrite amounts (pink triangles), which both increase with temperature. A possible reason for this volume expansion, as mentioned by Santofimia et al. [2], could be the bainite transformation. At 500 °C, however, it doesn’t follow the bainitic ferrite fraction; instead, slight contraction or almost no change occurs in comparison with other samples. This might be due to the following reasons. The first reason could be the result of an initial contraction that took place to compensate for a positive deviation in the thermal path when it reached the partitioning temperature. This deviation also appeared in other samples, but it can be ignored because their holding times are substantially longer and their effect on dilation is negligible. A second possible reason is that the carbon partitioning from martensite into austenite is controlled by the constrained carbon equilibrium (CCE) criterion, so there would not have to be any increase in the volume expansion if only this mechanism is activated [2].

The FWHM results for the austenite peaks at 500 °C show an increase in austenite carbon content homogenization, which is related to the equalization of the carbon in the austenite by diffusion from carbon-rich parts to areas with lower carbon content (from $\gamma_{HC}$ to $\gamma_{LC}$). In a study [6] examining austempering at different temperatures, the $\gamma_{HC}$ carbon content decreased from 1.30 to 1.14 wt.% and the $\gamma_{LC}$ increased from 0.22 to 0.34 wt.% after 300 s of austempering at 400 °C.

### 3.6. High- and Low-Carbon Austenite

Figure 6 illustrates a comparison of the X-ray diffraction line profile of the austenite $\gamma_{200}$ peak during partitioning at 280 °C and 500 °C. The evolution of the $\gamma_{200}$ peak for pure bainitic ferrite transformation at 280 °C is shown in Figure 6a. The austenite peak can be fitted to homogenous austenite at first, so no $\gamma_{HC}$ peak is present, but it then declines, whereas a $\gamma_{HC}$ peak grows somewhat with time. This is consistent with observations by Guo et al. [6] that the amount of filmy austenite between bainitic ferrite platelets increases as the bainite volume fraction increases during austempering. An asymmetrical broadening at a lower angle (4.45°) was found without any shift of the peak related to $\gamma_{LC}$, which shows a bimodal distribution of carbon in austenite. The overlapping of two Gaussian profiles may be used to describe the irregular peak form of austenite during the partitioning process. This implies that the austenite phase is divided into two states, each with its own set of lattice parameters, 3.618 and 3.660 nm. Using these lattice parameters, the carbon concentrations are determined to be 0.6 and 1.9 wt.%, respectively. Similar calculations were reported by Guo et al. [6] for a low-carbon steel, finding concentrations of 0.58 and 1.12 wt.% for $\gamma_{LC}$ and $\gamma_{HC}$, respectively. In contrast, the evolution of the $\gamma_{200}$ peak at 500 °C is depicted in Figure 6b, in which supersaturated martensite partitioning is prominent. The austenite peak can be fitted to a homogeneous austenite phase during the entire partitioning stage. The peak merely shifts to lower angles, suggesting that the mean lattice parameter of the austenite rises without any evidence of bimodal structure.

### 3.7. Effect of Partitioning Treatment on the Final Microstructure

The effect of the different partitioning treatments on the M_s_ temperature during the second quenching is illustrated in Figure 7a. It can be observed that in contrast with the general effect of carbon on M_s_ [57], the M_s_ temperature is not decreased by increasing the mean carbon amount partitioned into austenite. However, the higher amount of retained austenite at lower temperature Figure 7b, considering the bimodal austenite peaks (see Section 3.6), shows the higher stability of lath austenite formed during bainite formation at 280 °C. 

In addition to carbon partitioning from martensite to austenite, the bainite formation during the partitioning stage was another active mechanism for austenite carbon enrichment. When the partitioning temperature was increased, the dominant austenite enrichment mechanism changed from bainitic ferrite transformation to carbon partitioning from supersaturated martensite. It has been shown that partitioning at 280 °C for 900 s produces the highest percentage of retained austenite (almost 26 wt.%). 

Moreover, in comparison with a sample quenched directly to room temperature without partitioning treatment, the M_s_ temperature is significantly reduced. For the examined steel, the M_s_ of the sample in this condition is 183 °C and the retained austenite fraction is about 16 wt.%. The M_s_ temperature of the present steel, which is substantially lower than that of low-carbon steels, explains its high amount of retained austenite.

As shown in the introduction, a carbon threshold of about 0.7–1 wt.% is required to stabilize austenite at ambient temperature [7]. In the performed experiments, mainly due to the high cooling rate, the initial martensite fraction is about 0.11, which is much lower than predicted by the K–M equation or reported in previous observations about the same steel with a lower cooling rate. As a result, the amount of carbon that can be partitioned from martensite to austenite is limited. Additionally, as shown in Figure 5, by increasing partitioning temperature, the dominant austenite carbon enrichment mechanism changes from bainitic ferrite transformation to carbon partitioning from supersaturated martensite. Consequently, as Figure 7 illustrates, stabilization of retained austenite when the latter mechanism is dominant leads to a lower retained austenite fraction.

Based on the above discussion, the best approach for carbon enrichment in austenite with a low amount of pre-existing martensite seems to be via bainitic ferrite transformation by partitioning at lower temperatures, where more bainitic ferrite will be formed.

XRD analysis of the specimens showed that the microconstituents found in the final microstructure can be listed as follows: tempered martensite formed during the partitioning stage; bainitic ferrite and high-carbon austenite formed during partitioning; and fresh martensite which has been slightly tempered (during cooling below the M_s_ in the final quench). Additionally, to illustrate these constituents graphically, an optical micrograph of the specimen partitioned at 500 °C for 20 s and tint-etched using the Lepera method [58] is shown as one example in Figure 8. With this method of etching, (i) tempered martensite is shown in a blue scheme color [59], (ii) slightly tempered fresh martensite and ferritic bainite are shown in beige/light brown colors [60], and (iii) retained austenite remains white.

### 3.8. Influence of Prior Martensite on Bainite Transformation

Normally, the bainite reaction takes a long time to achieve a steady state in low-temperature austempering conditions, especially in the case of high-carbon high-silicon steels. According to Guo et al. [33], the incubation time required for bainite initiation can be thousands of seconds during austempering. Pre-existing martensite, on the other hand, can reduce the required incubation time by 30%, according to Ko and Cottrell [61]. Recent research has also demonstrated that preliminary martensite has the effect of accelerating bainite transformation [33,34,35,36].

Figure 9 shows the evolution of one of the major peaks of ferrite ($\alpha_{211}$) using in-situ HEXRD for two states, with and without partial quenching before isothermal holding at 280 °C in the Q&P and austempering processes, respectively (as shown in Figure 1). It can be seen that the prior martensite accelerates the subsequent low-temperature bainite transformation. The austempered sample exhibits no ferrite peaks after 900 s; only a slight bump of the $\alpha_{211}$ peak is apparent after 30 min of soaking. In contrast, the Q&P sample shows a reasonable increase for all peaks (as an example, the $\alpha_{211}$ peak is shown in Figure 9a). Figure 5a demonstrates that around 8% of bainitic ferrite is formed during the partitioning stage in the Q&P process. 

The accelerated phase transformation phenomenon could be explained by a change in the conditions for nucleation and growth of the bainite isothermal transformation, as well as by the lower energy requirement for heterogeneous nucleation as a result of increased dislocation density and stress fields caused by the martensite phase transformation expansion [33].

## 4. Conclusions

This study examines the dynamic microstructural changes occurring during heat treatment of Fe-0.6C-1.25Mn-1.6Si-1.75Cr-0.15Mo-0.12V (in wt.%) steel via an in-situ study by HEXRD and high-resolution dilatometry. Three Q&P schemes with different partitioning temperatures (280, 400 and 500 °C) were performed after full austenitizing at 910 °C and initial quench to 165 °C, and in addition one austempering process at 280 °C was applied. The main conclusions are as follows:In addition to carbon partitioning from martensite to austenite, the bainite formation during the partitioning stage was another active mechanism for austenite carbon enrichment. When the partitioning temperature was increased, the dominant carbon enrichment mechanism in the austenite changed from bainitic ferrite transformation to carbon partitioning from supersaturated martensite. It has been shown that partitioning at 280 °C for 900 s produces the highest percentage of retained austenite (almost 26 wt.%).In comparison with austempering heat treatment with no prior martensite, the presence of initial martensite in the Q&P microstructure accelerates the subsequent bainitic transformation during the partitioning stage.The FWHM results measured from $\gamma_{200}$ and $\gamma_{202}$ peaks during partitioning at 280 °C showed a positive slope, indicating heterogeneity in the austenite carbon content when bainitic ferrite is formed in the microstructure (this occurs especially at lower partitioning temperatures). The lattice parameters in the austenite varied from 3.618 to 3.660 nm, showing that the carbon concentration in the austenite was between 0.6 and 1.9 wt.%.At temperatures below 400 °C, a new bct phase was detected. Tetragonality of this phase was much higher than for ferrite (bcc structure), indicating that this phase could be tetragonal ferrite, which would be consistent with the symmetry of ferrite in equilibrium with austenite not being cubic when first formed, since it is supersaturated with carbon.The FWHM results for the austenite peaks at 500 °C indicate a more homogeneous distribution of carbon. This phenomenon is related to carbon diffusion in the austenite and an equalization of the carbon concentration in the austenitic grains.The dilation behavior of the samples shows a significant correlation with bainitic ferrite fraction as a function of increasing temperature. However, at 500 °C, when the bainitic ferrite fraction is minor and carbon partitioning from supersaturated martensite into austenite is the dominant mechanism, no volume expansion occurs.

## Figures and Tables

**Figure 1 materials-16-01557-f001:**
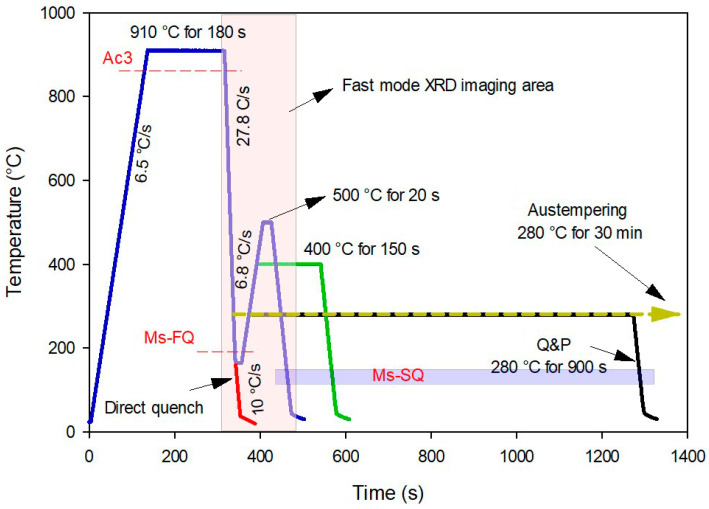
Heat treatments (austempering, direct quenching, and quenching & partitioning) were applied in this study. FQ and SQ represent the first and second quench, respectively.

**Figure 2 materials-16-01557-f002:**
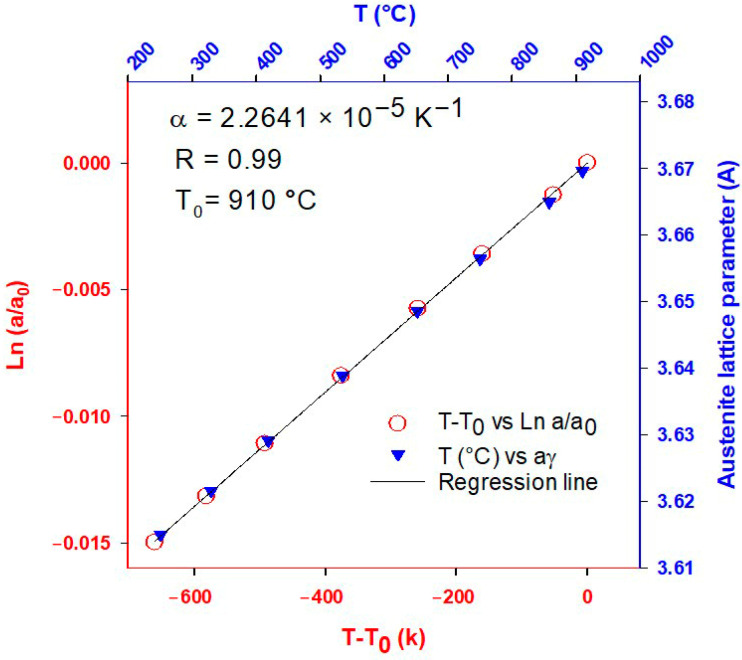
The evolution of the austenite lattice parameter, assuming that the alloy is fully austenitic in the temperature range. The mean slope is used to measure the thermal expansion parameter.

**Figure 3 materials-16-01557-f003:**
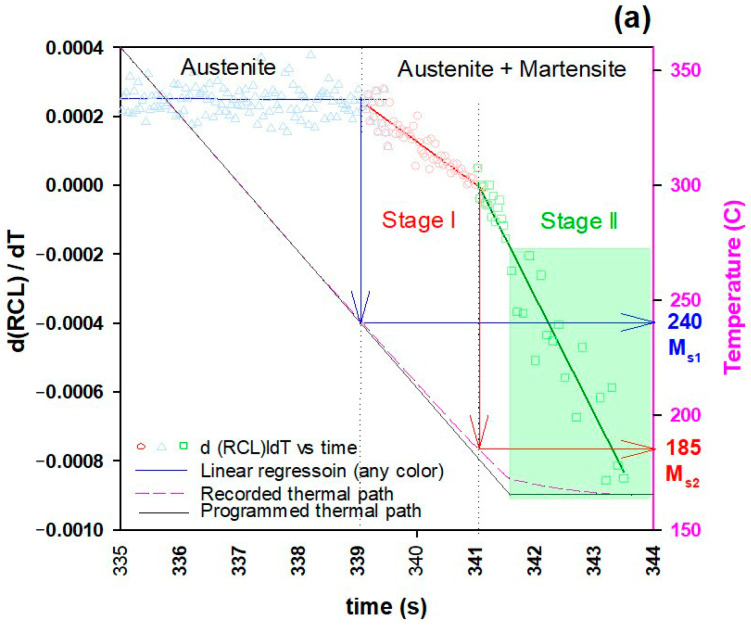
Dilation and XRD evolutions of the samples during the first quench to 165 °C: (**a**) relative change in length vs. temperature; (**b**) development of the different intensities of the lattice planes in austenite and martensite (colors from red to dark green represent highest to lowest intensities, respectively).

**Figure 4 materials-16-01557-f004:**
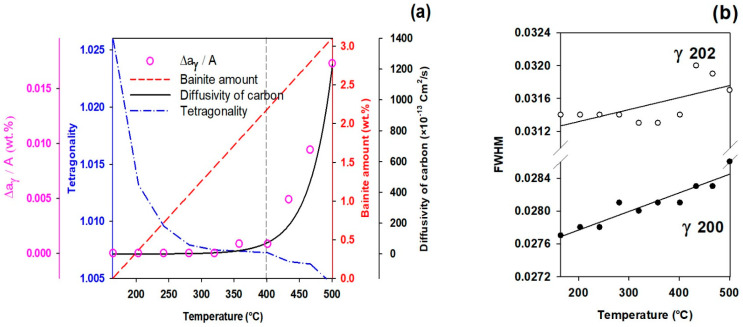
(**a**) Diffusivity of carbon into austenite, change in austenite carbon content as a function of $\frac{\Delta a_{\gamma }}{A}$, percentage of bainitic ferrite and carbon depletion from bct phase during heating from the quenching temperature (165 °C) to the partitioning temperatures; (**b**) Full width at half maximum peak values (FWHM) evolution of two austenite peaks (202 and 200) showing heterogeneity during reheating stage.

**Figure 5 materials-16-01557-f005:**
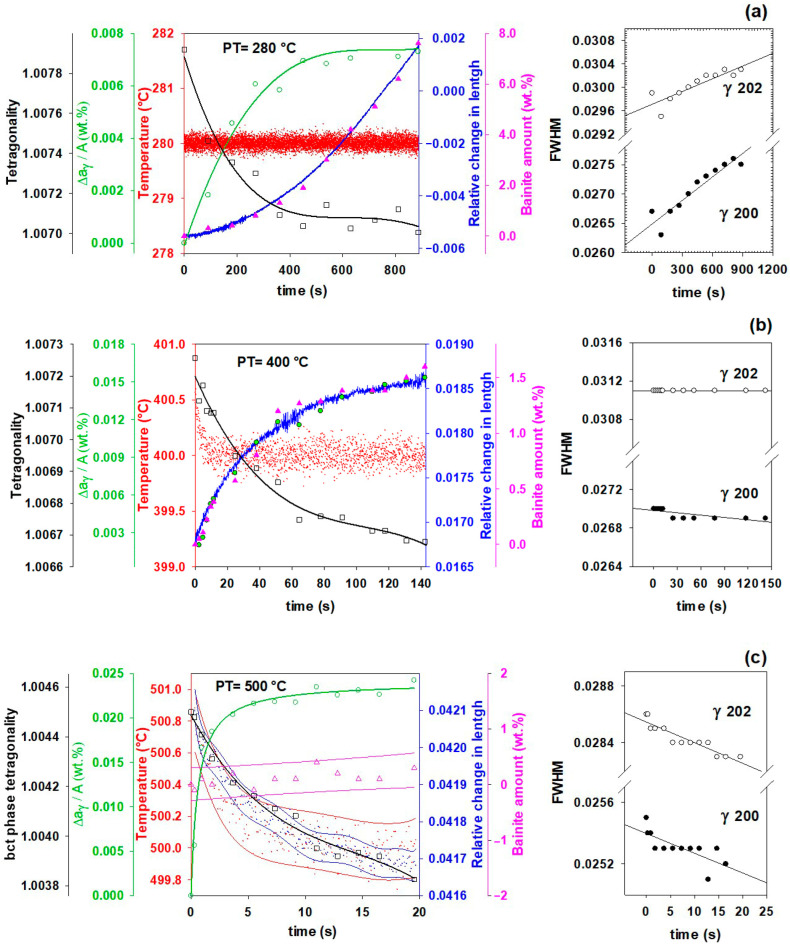
Microstructural evolution of bainitic ferrite and austenite measured by XRD analysis during partitioning at (**a**) 280 °C, (**b**) 400 °C, and (**c**) 500 °C. (For interpretation of the colors in this figure legend, see the online version of this article).

**Figure 6 materials-16-01557-f006:**
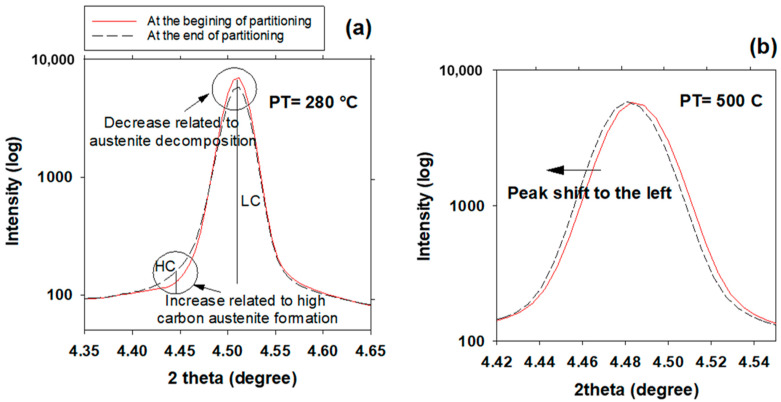
X-ray diffraction line profile of austenite $\gamma_{200}$ during partitioning at (**a**) 280 °C and (**b**) 500 °C, showing bimodal and homogenous austenite phase compositions, respectively.

**Figure 7 materials-16-01557-f007:**
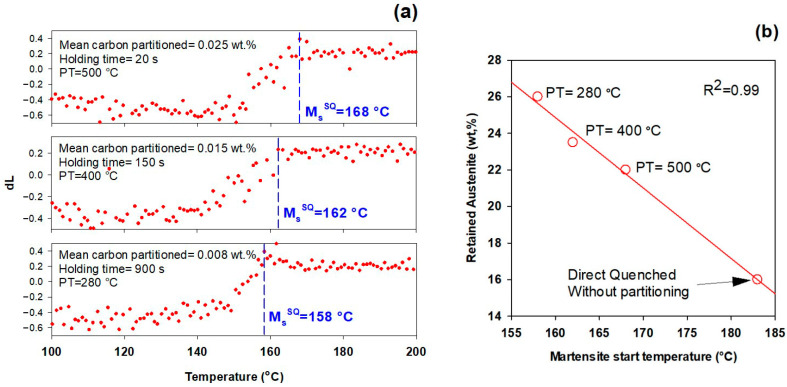
(**a**) Effect of partitioning treatment on martensite start temperature (M_s_); (**b**) Effect of M_s_ temperature on retained austenite fraction.

**Figure 8 materials-16-01557-f008:**
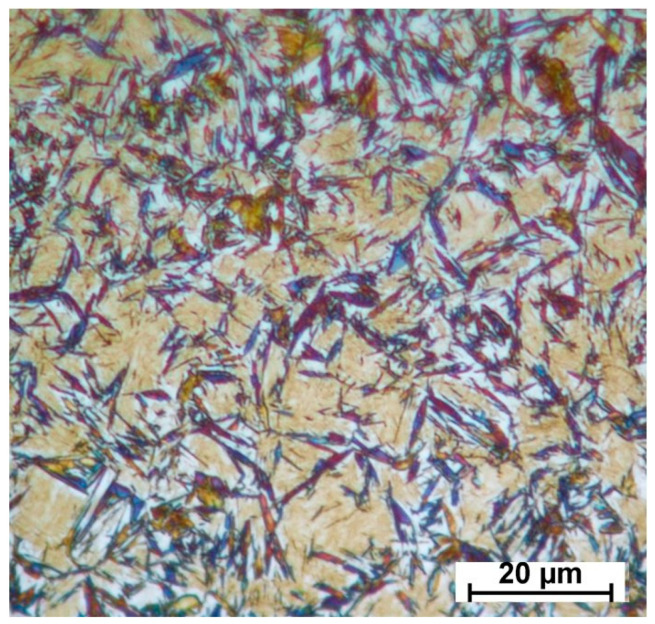
Optical micrograph of the specimen partitioned at 280 °C for 900 s, etched with LePera etchant [58]. (For interpretation of the colors in this figure legend, see the online version of this article).

**Figure 9 materials-16-01557-f009:**
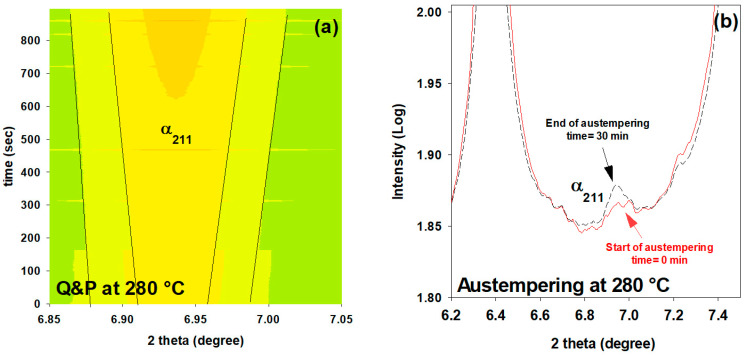
Evolution of ferrite 211 peak during isothermal holding at 280 °C for two states: (**a**) with partial quenching (Q&P treatment) and (**b**) without partial quenching (austempering treatment). In figure (**a**), intensity is logarithmic and colors from green to dark yellow show the lowest (1.70) and the highest (2.20) intensity, respectively. (For interpretation of the colors in this figure legend, see the online version of this article).

## Data Availability

Not applicable.

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
