# Peer review of "Kinetics of Carbon Enrichment in Austenite during Partitioning Stage Studied via In-Situ Synchrotron XRD"

_materials, 2023, doi:10.3390/ma16041557_

Round 1

Reviewer 1 Report

this paper carried out an interesting study on carbon enrichment in austenite during partitioning stage. the results and analysis are sufficient and reasonable. following comments should be addressed before being accepted.

1, double check the size of hollow cylinders, in page 3 line 101;

2, explain "the martensite exhibits carbon depletion as the tetragonality decreased by quenching temperature, from 1.030 at 183 °C to 1.026 at 165 °C", without carbon enrichment in austenit.

3, page 11, line 360: why homogeneity were increased?

4, the discussion and explanation on increased Ms temperature with increased partitioned carbon amount, in part 3.7 should be more clear. 

Author Response

Response to Reviewers’ Comments

Manuscript ID:  materials-2159137

Title: Kinetics of carbon enrichment in austenite during partitioning stage studied via in-situ synchrotron XRD

We sincerely acknowledge our deep sense of gratitude to you for giving valuable time to assess the manuscript. We are also very much thankful to the reviewers for their valuable comments and constructive suggestions, which are being implemented/corrected for the necessary improvement of the manuscript.

The changes/additions are highlighted in the revised manuscript with yellow color.

Response to Reviewer #1

Reviewer #1:  

1, double check the size of hollow cylinders, in page 3 line 101;

Response: Thank you for your comment. We have checked the sizes and our writing and there is a small mistake. The samples are machined into hollow cylinders with 4 mm external diameter, with 1 mm wall thickness and 10 mm length (L0) in order to let the quenching gas flow through the sample and minimizing the temperature gradient over the thickness (with other words had the center hole a diameter of 2 mm).

2, explain "the martensite exhibits carbon depletion as the tetragonality decreased by quenching temperature, from 1.030 at 183 °C to 1.026 at 165 °C", without carbon enrichment in austenit.

Response: Thank you for your comment. As described in section; 3.4.2 Tetragonal bct phase; changing in tetragonality has a direct relationship with carbon content (), so changing in tetragonality from 1.3 to 1.26 shows the carbon depletion occurrence during cooling from Ms temperature (183°C) to the quench stop temperature (165 ℃). However, no change of the austenite peaks was observed in this temperature range which means that the time at temperature, was not yet enough to let the carbon diffuse into austenite. We should consider that the diffusivity of carbon in ferrite/martensite is around 100 times faster than in austenite, also the amount of carbon that is supposed to diffuse in this case out from the 10% martensite to the 90% of austenite in the structure, is a very small part and that could not be detected even by synchrotron XRD.

3, page 11, line 360: why homogeneity were increased?

Response: Thanks for your point. As showed in Figure 6, there is non-homogeneity in retained austenite of samples treated at lower temperature (280 °C) which resulted in 2 peaks of high carbon austenite (g HC) and low carbon austenite (g LC), so the sentence is modified and a clarification is added to section 3.7.

Action: Text from line 358 to 370 is modified, to the following:

The effect of the different partitioning treatments on Ms temperature during the second quenching is illustrated in Figure 7a. It can be observed that in contrast with the general effect of carbon on Ms [57], the Ms temperature is not decreasing by increasing the mean carbon amount partitioned into austenite. However, higher amount of retained austenite at lower temperature (Figure 7b) with considering the bimodal austenite peaks (see section 3.6) shows the higher stability of lath austenite formed during bainite formation at 280°C.

In addition to carbon partitioning from martensite to austenite, the bainite formation during the partitioning stage was another active mechanism for austenite carbon enrichment. When the partitioning temperature was increased, the dominant austenite enrichment mechanism changed from bainitic ferrite transformation to carbon partitioning from supersaturated martensite. It has been shown that partitioning at 280 °C for 900 s produces the highest percentage of retained austenite (almost 26 wt.%).

4, the discussion and explanation on increased Ms temperature with increased partitioned carbon amount, in part 3.7 should be more clear. 

Response: As can be seen in Figure 7a and 7b, the lower the partitioning temp the lower the second Ms temperature and higher the amount of retained austenite. However, the fraction of martensite after the first quench is the same for all conditions, so the other source of carbon which is ferritic bainite is playing the major role in this case. As shown in Figure 6, bimodal peaks of austenite are detected at 280°C partitioning, so it means that the morphology of the retained austenite laths formed between the bainite lathes is the reason for increasing the stability of the small amount of extra austenite that could be stabilized during portioning at 280°C. 

Action: As mentioned for previous question, the text from line 358 to 370 is modified.

Reviewer 2 Report

See file

Author Response

Response to Reviewers’ Comments

Manuscript ID:  materials-2159137

Title: Kinetics of carbon enrichment in austenite during partitioning stage studied via in-situ synchrotron XRD

We sincerely acknowledge our deep sense of gratitude to you for giving valuable time to assess the manuscript. We are also very much thankful to the reviewers for their valuable comments and constructive suggestions, which are being implemented/corrected for the necessary improvement of the manuscript.

The changes/additions are highlighted in the revised manuscript with yellow color.

Response to Reviewer #2

Reviewer #2:

1, the work is very interesting and informative with regards to the Q&P process and the difference between high and low temperature partitioning treatments and confirms other studies.

However, it is a shame that this work is not combined with the relationship of these different

structures to the mechanical properties, yield strength and ductility as the ultimate aim of this

work is to improve the properties of steel.

Response: Thank you for your comment. The authors previously studied the mechanical properties of similar Q&P conditions on 06CV steel and results are published in the following paper which is Reference [59] in this manuscript.

-Forouzan, F., Borasi, L., Vuorinen, E., & Mücklich, F. (2019). Process control maps to design an ultra-high strength-ductile steel. Materials Science and Technology, 35(10), 1173-1184.

2, in the conclusions it would be helpful if you could list what is important to achieve the best

properties from the literature and on the basis of this which of the treatments you have given

will most likely achieve this.

Response: Authors believe that what is mentioned under the first bullet point of conclusions, regarding the change of the dominant phenomena to bainite formation at low temperature partitioning condition which is resulted to the highest retained austenite is the most important fact to a select the conditions of the treatment for other high silicon steels depending on their Ms temperature.

3, however, although I am sure your general conclusions and the experimental work and

interpretation are fine I do feel that you need to make what you have done clearer. For example

on p6. First quench (initial martensitic state) is this quenching to 165oC or direct quenching to

room temperature or quenching to 165oC and then partitioning at 500C and then quenching to

room temperature . I understand that Ms1 is the direct Ms1 and when you partition, the carbon

content of the austenite is increased so you have another Ms2 at a lower temperature. Again

Fig.3 it is not clear what exactly you are doing. It seems you are scanning the planes from 2theta

3.85 to 9.1o starting at around 183oC when the first martensite peaks appear and the austenite

111 plane is dominant but the temperature does not seem to go down systematically with

diffaction angle.

Can you please make the paper easier to understand because it is a very good paper.

Response: This section (p.6 line 202- 206) is modified to make it clearer. Thanks for the comment.

Figure 3 depicts the evolution of the γ→ά transformation of the as-quenched specimen. It is interesting that the slope of dilation ( vs. time shows 2 times changes in the slope, indicating the first Ms temperature (=240 °C) and the second Ms temperature () at around 185°C. Although XRD patterns do not show any martensite peaks in stage I, XRD data confirms well the second Ms temperature ().

4, there are a lot of minor corrections you must make, to make it easier reading.

Although the use of acronyms saves time, they are only good if readers are very familiar with

them.

On P5 line 176 put coefficient of thermal expansion (CTE): Done

P.7 line 239 start sentence. The full width half maximum peak values (FWHM) Done

Also put in what APT and EBSD stand for in line 60 and line 104, respectively. Done

P.1, Line 20. Although at higher temperatures. The word, "Although" is confusing better to put

just put "At higher temperatures..". Done

P.1, Line 42. However, the latter not later. Done

P.2, Line 44 should it not be 1.12 and 0.58 wt.%, respectively? Done

P.5 lines 189-194 are not understandable probably through poor English. Could you rewrite this

so that it is easier to read.

However, the thermal expansion effect at elevated temperatures must be excluded, and this can be done by the expression proposed by Denand et al. [41], Equation 4.

                                                                        Equation 4

Where  is the total carbon concentration change in austenite,  is the difference in lattice parameter of austenite without thermal contribution over time at the same temperature,  is the nominal carbon content of the steel and ? is the constant parameter of 0.033 wt.% extracted from Dyson & Holmes equation.

P.6. line 197. First quench approx.165oC ? (initial martensite state)

Response: The temperature 165 °C is added to the figure caption:

Figure 3. Dilation and XRD evolutions of the samples during the first quench to 165 ℃. a) Relative change in length vs. temperature, b) Development of the different intensity of the lattice planes in austenite and martensite (colors from red to dark green represent highest to lowest intensities, respectively).

P6. Fig. 3b is not clear. Are you trying to show that when you quench to 165oC (first quench),

that only when the temperature is 183oC is there martensite in the structure and the Ms2

temperature must be close to this?

Yes, according to this graph, when the temperature is reduced, martensite peaks appear gradually. The first doublet peaks that show up at 183.1 °C are 101 and 110, indicating that this temperature is a good estimate of .

Moreover, when temperature is almost 170 ℃ mostly all the major martensite peaks had appeared, the following table should make it easier to understand.

Note: Numbers in the table show when a peak appeared in seconds in the experiment.

Martensite peaks

101

110

002

200

310

130

112

211

   183.1 C

183.1 C

341

341

175.4 C

341

341

341.5

341.5

171.5 C

341

341

341.5

341.5

341.8

341.8

169.9 C

341

341

341.5

341.5

341.8

341.8

342

342

P.8 Fig.4b. Along Y axis put in Full width at half minimum peak values (FWHM) : Done

P.9, line 295. “On the contrary—better “In contrast, at 500oC: Done

Reviewer 3 Report

The article highlights peculiarities of the microstructural evolution and corresponding mechanisms occurring during different stages of quenching and partitioning conducted on 0.6C-1.5Si steel by in-situ High Energy X-Ray Diffraction and high-resolution dilatometry methods. The authors used the modern equipment for visualization and assistance in the interpretation of the obtained results. The authors should pay attention to the fact that the TRIP effect is closely related to the content of only blocky-like retained austenite.

The article is interesting, but a number of shortcomings need to be corrected:

1.     Please state manufacturer, city and country from where equipment has been sourced. This have to be done for each equipment, software, material and chemical in the paper.

2.     The authors should note that not only the exposure time affects the structure of steel during treatment, but also the cooling rate. (For example, https://doi.org/10.1007/s11003-019-00263-6)

3.     References to Figures 6 and 10 are given after Figures 3 and 6 respectively. This should be fixed.

4.     The font size in Figures 5а and 6 should be increased. Also Fig. 6 is of poor quality and is cropped.

5.     Table 2 is mentioned in the text of the article (Line 411). This is obviously a typo that needs to be corrected.

6.     The authors would do well to separate the content of film-like RA and blocky-like RA, since the TRIP effect is closely related to the content of only blocky-like retained austenite (over the past year, a large number of publications are devoted to the problems of film-like RA and blocky-like RA).

Author Response

Response to Reviewers’ Comments

Manuscript ID:  materials-2159137

Title: Kinetics of carbon enrichment in austenite during partitioning stage studied via in-situ synchrotron XRD

We sincerely acknowledge our deep sense of gratitude to you for giving valuable time to assess the manuscript. We are also very much thankful to the reviewers for their valuable comments and constructive suggestions, which are being implemented/corrected for the necessary improvement of the manuscript.

The changes/additions are highlighted in the revised manuscript with yellow color.

Response to Reviewer #3

Reviewer #3:

The article highlights peculiarities of the microstructural evolution and corresponding mechanisms occurring during different stages of quenching and partitioning conducted on 0.6C-1.5Si steel by in-situ High Energy X-Ray Diffraction and high-resolution dilatometry methods. The authors used the modern equipment for visualization and assistance in the interpretation of the obtained results. The authors should pay attention to the fact that the TRIP effect is closely related to the content of only blocky-like retained austenite.

The article is interesting, but a number of shortcomings need to be corrected:

  1. Please state manufacturer, city and country from where equipment has been sourced. This have to be done for each equipment, software, material and chemical in the paper.

Response: Thank you for your comment, the following changes have been applied in the materials and method section text:

  1. produced by ASCOMETAL, France                     line 99
  2. by LaB6 powder (SRM 660C NIST, USA) line 123
  • by a Perkin Elmer XRD1621, USA                     line 124
  1. from TA Instruments, USA                     line 134
  2. The authors should note that not only the exposure time affects the structure of steel during treatment, but also the cooling rate. (For example, https://doi.org/10.1007/s11003-019-00263-6)

Response: The exposure time that is mentioned in the article is regarding to the time that xrd pattern is recorded during this work which was 3.2 s for heating and 0.3 s for the quenching section and the beginning of partitioning. During this time the procedure of heat treatment is not stopped only it in noted to consider that the effect of changing in the microstructure over this period could be affecting the pattern to be different a little bit from low 2theta to the high 2thetas.

  1. References to Figures 6 and 10 are given after Figures 3 and 6 respectively. This should be fixed.

Response: Unfortunately, we don’t understand what you mean exactly in this question. We have checked the whole manuscript and all figures are only referred just after it is added. Also, this article has totally 9 figures so there is no figure 10.

  1. The font size in Figures 5а and 6 should be increased. Also Fig. 6 is of poor quality and is cropped.

Response: Thanks for your comment. Figures 5 and 6 were rebuilt according to the reviewer’s point.                                                             

  1. Table 2 is mentioned in the text of the article (Line 411). This is obviously a typo that needs to be corrected.

Response: Thanks. Table 2 and lines 415 and 416 are deleted.       

  1. The authors would do well to separate the content of film-like RA and blocky-like RA, since the TRIP effect is closely related to the content of only blocky-like retained austenite (over the past year, a large number of publications are devoted to the problems of film-like RA and blocky-like RA).

Response: Thank you for your comment. Authors do agree with your point regarding the difference in the stability and trip effect of film-like vs blocky-like RA, however this work is only focused on the High energy XRD results, so it is impossible to distinguish the morphology of one phase by this method. But the text in section 3.7 line 358 to 364 is modified and the formation of lath RA is added regarding the reasoning of the higher stability of RA at lower partitioning temperature, as below line 366-371:

In addition to carbon partitioning from martensite to austenite, the bainite formation during the partitioning stage was another active mechanism for austenite carbon enrichment. When the partitioning temperature was increased, the dominant austenite enrichment mechanism changed from bainitic ferrite transformation to carbon partitioning from supersaturated martensite. It has been shown that partitioning at 280 °C for 900 s produces the highest percentage of retained austenite (almost 26 wt.%).

Round 2

Reviewer 3 Report

The authors took into account almost all comments of the reviewer and made appropriate corrections to the manuscript.

However, a number of shortcomings need to be corrected:

1.     References to Figures 3 are given after Figures 3. This should be fixed.

Author Response

Response to Reviewer Comments

Manuscript ID:  materials-2159137

Title: Kinetics of carbon enrichment in austenite during partitioning stage studied via in-situ synchrotron XRD

We are very thankful for the comments given by the reviewers and we have done our best to improve the article according to the comments and suggestions by the reviewers.

The changes/additions in the revised manuscript have been introduced by marking them by using the “Track Changes” function.  

Response to Reviewer #3

The article is interesting, but a number of shortcomings need to be corrected:

  1. References to Figures 3 are given after Figure 3. This should be fixed.

We apologize for not have corrected this mistake already in our first trial in answering of the reviewers comments. We have now made a rearrangement of the text according to the order presented below. The first sentence is added in order to give a better introduction to this section of the article (this is marked with a yellow color below).

  • First quench (initial martensite state)

The austenitisation was performed at 910 ℃ and the first quenching treatment was  stopped at 165 ℃. Figure 3 depicts the evolution of the γ→ά transformation of the as-quenched specimen. It is interesting that the slope of dilation ( vs. time shows 2 times changes in the slope, indicating the first Ms temperature (=240 °C) and the second Ms temperature () at around 185°C. Although XRD patterns do not show any martensite peaks in stage I, XRD data confirms well the second Ms temperature (). Figure 3b represents a logarithmic contour graph of the XRD patterns versus time. According to this graph, when the temperature is reduced, martensite peaks appear gradually. The first doublet peaks that show up at 183.1 °C are 101 and 110, indicating that this temperature is a good estimate of .

The authors [49] noted that micro-segregation generates bands of enriched and deficient Mn–Cr areas, which affect the Ms temperature of the bands in a separate experiment on the same steel (bands with higher amounts of Cr-Mn result in lower Ms temperature, and vice versa). So, it is reasonable to conclude that partial segregation has affected the first quench based on the foregoing observations and previous research on the studied steel. At =240 °C, the minor bands, which are depleted Mn–Cr areas, begin to transform into martensite, however, the percentage must be below the synchrotron detection limit. Afterward, the major enriched Mn–Cr regions start transforming at =183 °C. Consequently, as the undercooling is around 20 °C, only 11.5 wt.% martensite is formed.

Figure 3. Dilation and XRD evolutions of the samples during the first quench to 165 ℃. a) Relative change in length vs. temperature, b) Development of the different intensity of the lattice planes in austenite and martensite (colors from red to dark green represent highest to lowest intensities, respectively).

Moreover, at this temperature range (183 à165 ℃), the martensite exhibits carbon depletion as the tetragonality decreased by quenching temperature, from 1.030 at 183 °C to 1.026 at 165 °C. Though, estimation of the average carbon atom diffusion in austenite by Equation 5, results in a radial distance (r) of 0.4 nm that a carbon atom would move after 13 seconds of holding time at 165 °C, which cannot considerably change the carbon enrichment in austenite. This also explains the constant value of the mean austenite carbon content measured by XRD Rietveld refinement analysis during this stage.

                                                                                                                                                                                                                                                                 Equation 5

D in Equation 5 is the carbon diffusivity which is defined by Equation 6 and t is time in seconds.

                                                                                                                                          Equation 6

In Equation 6, Rc is the constant (1.987 ), T is the temperature in Kelvin and C is the nominal carbon content of the steel (in wt.%) [50].
